# Complementing Self-Consistency with Cross-Model Disagreement for Uncertainty Quantification

**Kimia Hamidieh[1], Veronika Thost[2], Walter Gerych[1],**
**Mikhail Yurochkin[3]\*, Marzyeh Ghassemi[1]**

[1]MIT, [2] MIT-IBM Watson AI Lab, [3] IFM MBZUAI
{hamidieh, wgerych, mghassem}@mit.edu

## Abstract

Large language models (LLMs) often produce confident yet incorrect responses, and uncertainty quantification is one potential solution to more robust usage. Recent works routinely rely on self-consistency to estimate aleatoric uncertainty (AU), yet this proxy collapses when models are overconfident and produce the same incorrect answer across samples. We analyze this regime and show that cross-model semantic disagreement is higher on incorrect answers precisely when AU is low. Motivated by this, we introduce an epistemic uncertainty (EU) term that operates in the black-box access setting: EU uses only generated text from a small, scale-matched ensemble and is computed as the gap between inter-model and intra-model sequence-semantic similarity. We then define total uncertainty (TU) as the sum of AU and EU. In a comprehensive study across five 7–9B instruction-tuned models and ten long-form tasks, TU improves ranking calibration and selective abstention relative to AU, and EU reliably flags confident failures where AU is low. We further characterize when EU is most useful via agreement and complementarity diagnostics.

## 1 Introduction

Reliable uncertainty estimates are a prerequisite for deploying large language models (LLMs) in high-stakes domains [5]. Many existing approaches for LLM uncertainty estimation are based on model's *self*-confidence [65, 61, 55], such as by measuring response consistency under sampling [31, 39, 50, 2] or querying for a verbalized uncertainty score [38]. These metrics capture how internally confident a model is in its prediction – a notion of *predictive aleatoric uncertainty* (AU). But this leaves an important question unanswered: how confident should *we* be in the model? A model might be *confident but wrong*, such as responding with the same incorrect answer with high probability (see Figure 1). In these cases, methods that rely on self-consistency can fail [30]. To address this, we focus on estimating *epistemic uncertainty* (EU) – uncertainty in our *choice* of model – which better reflects whether a model's confidence is trustworthy for a given input.

Estimating EU requires evaluating a distribution of plausible models, which is prohibitively costly for LLMs, as training even one additional model adds significant overhead [29, 9]. Recent shortcuts approximate EU in logit space [43], inject Bayesian noise during decoding [41, 16], or rely on verifier–model disagreement [66], but each imposes strong task or architecture-specific assumptions. We instead capitalize on the ecosystem of open-weight LLMs: sampling responses from a small, scale-matched ensemble lets us estimate EU directly from cross-model semantic disagreement, without additional training. While prior work has shown that LLM ensembles can improve accuracy [42, 11, 6, 54, 22], their use for uncertainty quantification has not been systematically explored.

---

\*Work done while at MIT-IBM AI Lab

39th Conference on Neural Information Processing Systems Workshop (NeurIPS 2025).

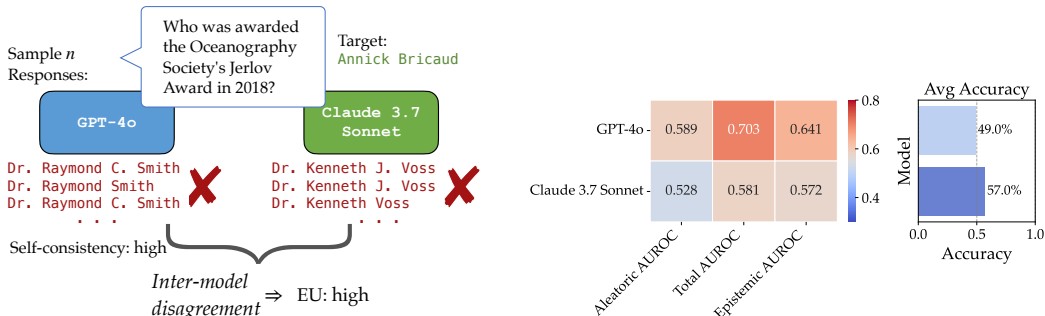

Figure 1: (a) Two models confidently produce distinct, incorrect answers to a factual question, which results in low intra-model variability (AU) but high semantic disagreement across models (EU). (b) Total uncertainty (TU = AU + EU) effectively improves uncertainty calibration with correcness in terms of AUROC on `SimpleQA`.

By enabling scalable estimation of EU from model outputs alone, we can combine it with AU to obtain a more robust measure of uncertainty, Total Uncertainty (TU). These two forms of uncertainty are complementary; AU reflects variability in a model's own predictions, while EU measures divergence from other plausible models [53]. Together, they allow TU to account for both internal inconsistency and external disagreement.

We evaluate EU, AU and TU on two standard axes: ranking-based calibration (via AUROC) and selective prediction (via abstention under uncertainty thresholds), across a range of models and generation tasks. We conduct comprehensive experiments across five 7–9B parameter Instruction-tuned models [67, 23, 18, 17], on *ten* long-form generation tasks spanning QA, summarization, translation, and math reasoning [26, 68, 47]. We also repeat these experiments for API models such as `GPT-4o` [21] and `Claude 3.7 Sonnet` [7] on `SimpleQA` [64]. Our contributions are as follows:

- We diagnose the failure mode of self-consistency as an aleatoric proxy: it often collapses on *confident errors*, or prompts where the model produces the same wrong answer repeatedly.
- We introduce an epistemic term based on cross-model semantic disagreement within a small, scale-matched ensemble, and show that it reliably identifies confident but incorrect responses.
- We conduct an extensive empirical study across ten long-form generation tasks and five reference models, and show that TU consistently outperforms AU in both AUROC and selective abstention.
- We show that the proposed EU is most informative in tasks with a unique correct answer, such as factual QA and translation.

## 2 Related Works

**Aleatoric Uncertainty in LLMs.** Existing approaches mainly focus on AU, which captures response inconsistency or input ambiguity. Recent surveys provide extensive reviews of these methods [65, 55, 61]. Typical strategies involve sampling multiple responses per prompt and analyzing their semantic consistency, often through clustering or entropy-based metrics [39]. In line with prior evaluations, we adopt degree-based semantic dispersion [39] as the AU baseline used throughout this paper.

**Bayesian-inspired EU Estimation.** A line of research employs Bayesian-inspired methods, such as adding noise to embeddings during generation to approximate uncertainty in model weights [41], sampling from a model with different temperatures [16], or leverage entropy from decoding from different hidden states as proxy for uncertainty, which provides a computationally efficient alternative to exhaustive sampling [15]. Training ensembles explicitly, such as LoRA-based methods [62], demonstrate improved uncertainty calibration but incur significant computational costs. Another work [43] calculates AU and EU on token level by considering the LLM logits as parameters of a Dirichlet distribution and by applying other UQ methods subsequently. Unlike approaches that require logits or hidden states, whereas our estimator operates only on generated text (Eq. 3), which enables application to black-box models.

**Prompt-Based and Verifier-Based EU Estimation.** Prior works in iterative prompting [1, 25] estimate epistemic uncertainty by iteratively querying the same model, adding previous responses to the later queries' prompts, and measuring probabilistic inconsistencies. However, these methods have limitations: gains over AU are mainly reported on multi-label data, with limited benefits on standard single-label QA [1], and some are evaluated only on synthetic data [25]. Xue et al. [66] utilizes one verifier LLM and shows that inter-model disagreement as a proxy for EU complements AU in cases where we reach the performance bounds of self-consistency. Their practical rule triggers

cross-consistency only at intermediate AU, whereas our evaluation, and [56], indicates that low AU is especially prone to hallucination and is best complemented by EU (Sec. 5.1). Prior works are mostly limited to special kinds of question-answering data, do not account for the impact of the vast LLM model space on EU, and do not fully explore interactions between AU and EU, gaps our work addresses explicitly.

**LLM Ensembles.** Our work builds upon classical uncertainty estimation, such as deep and dropout ensembles [35, 14] and is closely related to LLM ensemble applications [42]. In particular, various recent works focus on LLM collaborations [11], verifier LLMs [36], and sampling from multiple LLMs [6]. We study LLM ensembles from the viewpoint of uncertainty estimation.

## 3   Quantifying Predictive Uncertainty Using Response Similarity

Let $\omega$ be the particular parameterization of an LLM, and let $x$ be a prompt. Our goal is to quantify the predictive uncertainty of $\omega$ given $x$ as input. As is standard, we categorize predictive uncertainty into two components: AU and EU [20]. The aleatoric component captures the inherent unpredictability of the response to $x$ under the model $\omega$, while the epistemic component captures our uncertainty in $\omega$ being the correct parametrization to use when responding to input $x$. We define total predictive uncertainty additively as the sum of the aleatoric and epistemic uncertainties.

### 3.1   Aleatoric Uncertainty via Intra-Model Response Similarity

Many recent works have proposed techniques to measure the randomness in LLM responses [32, 39, 40]. These techniques typically focus on measures of *semantic* uncertainty, where uncertainty is defined as a function of how often an LLM produces semantically distinct outputs given the same input [32, 39]. In particular, [39] propose a measure equivalent to[2] the following:

$$U_{\text{aleatoric}}(x;\omega) = \mathbb{E}_{r_1^\omega \sim p(\cdot|x,\omega)} \mathbb{E}_{r_2^\omega \sim p(\cdot|x,\omega)} \big[1 - s(r_1^\omega, r_2^\omega)\big], \tag{1}$$

where $s(\cdot, \cdot)$ is a similarity metric for responses such as the cosine similarity in an embedding space.

In essence, Equation 1 corresponds to the expected similarity between two responses independently sampled from $p(\cdot|x,\omega)$, the response distribution of $\omega$ conditioned on $x$. If responses typically have the same semantic meaning as each other, meaning that the meaning of the response does not vary when resampled, then $U_{\text{aleatoric}}(x;\omega)$ will be close to 0, which means that there is little uncertainty in how $\omega$ will respond to $x$. If the model is likely to produce semantically distinct responses for the same input, then $U_{\text{aleatoric}}(x;\omega)$ will be high, which means $\omega$ has high uncertainty for $x$.

Equation 1 captures the inherent uncertainty in the response to $x$ given model $\omega$. However, $\omega$ may not be the optimal model to use for $x$, and Equation 1 fails to capture the inherent uncertainty that comes from choosing $\omega$ as our parameterization. There is thus a need to also capture the *epistemic* uncertainty that comes from our model choice.

### 3.2   Epistemic Uncertainty as Inter-Model Response Similarity

Let $\omega^*$ represent a hypothetical "ideal" model, such that $p(\cdot|x;\omega^*) = p(\cdot|x)$; the distribution of responses from $\omega^*$ equals the true response distribution. We can thus quantify the epistemic uncertainty of $\omega$ as a divergence between $\omega$ and $\omega^*$; e.g., $U_{\text{epistemic}}(x,\omega) = D(\omega \,||\, \omega^*)$ [53]. We define $D$ as follows:

$$D(\omega \,||\, \omega^*) = -\Big[ \underbrace{\mathbb{E}_{q_1^\omega \sim p(\cdot|x,\omega)} \mathbb{E}_{q_2^{\omega^*} \sim p(\cdot|x,\omega^*)} \big[s(q_1^\omega, q_2^{\omega^*})\big]}_{\text{cross-model similarity}} - \underbrace{\mathbb{E}_{r_1^\omega \sim p(\cdot|x,\omega)} \mathbb{E}_{r_2^\omega \sim p(\cdot|x,\omega)} \big[s(r_1^\omega, r_2^\omega)\big]}_{\text{self-similarity (1 - AU)}} \Big].$$
$$\tag{2}$$

In effect, $D(\omega \,||\, \omega^*)$ measures the difference between 1) the similarity of responses from $\omega$ and $\omega^*$ $\big(\mathbb{E}_{q_1^\omega \sim p(\cdot|x,\omega)} \mathbb{E}_{q_2^{\omega^*} \sim p(\cdot|x,\omega^*)} \big[s(q_1^\omega, q_2^{\omega^*})\big]\big)$ and 2) the self-similarity of $\omega$ $\big(\mathbb{E}_{r_1^\omega \sim p(\cdot|x,\omega)} \mathbb{E}_{r_2^\omega \sim p(\cdot|x,\omega)} \big[s(r_1^\omega, r_2^\omega)\big]\big)$. In the case where $\omega$ is optimal and equivalent to $\omega^*$, then $D(\omega \,||\, \omega^*)$ will be 0. When $\omega$ produces responses that are semantically diverse from the ideal model's responses even after accounting for the diversity due to $\omega$'s aleatoric uncertainty, then $D(\omega \,||\, \omega^*)$ will

---

[2]This is equivalent to $U_{Deg}$ in [39].

be high. In practice, we do not have access to the optimal model $\omega^*$. Instead, we can leverage a recent information-theoretic technique [53] and marginalize out $\omega^*$. Let $P_\Omega$ be a distribution over models such that $\mathbb{E}_{\tilde{\omega} \sim P_\Omega}\big[p(\cdot \mid x; \tilde{\omega})\big] = p(\cdot \mid x)$. We can thus replace $\omega^*$ in Equation 2 with an expectation over $P_\Omega$, and define $U_{\text{epistemic}}(x, \omega)$ as:

$$
\begin{aligned}
U_{\text{epistemic}}(x, \omega) = & - \mathbb{E}_{\tilde{\omega} \sim P_\Omega}\Big[\mathbb{E}_{q_1^\omega \sim p(\cdot|x,\omega)} \mathbb{E}_{q_2^{\tilde{\omega}} \sim p(\cdot|x,\tilde{\omega})}\big[s(q_1^\omega, q_2^{\tilde{\omega}})\big]\Big] \\
& + \mathbb{E}_{r_1^\omega \sim p(\cdot|x,\omega)} \mathbb{E}_{r_2^\omega \sim p(\cdot|x,\omega)}\big[s(r_1^\omega, r_2^\omega)\big].
\end{aligned}
\tag{3}
$$

When the average similarity in responses between $\omega$ and other sampled models matches the self-similarity of $\omega$'s responses, then the semantic distribution of $\omega$ matches the target distribution and the epistemic uncertainty is low. When there is a mismatch between the average similarity of $\omega$'s responses to the responses from the sampled models $\tilde{\omega}$ compared to $\omega$'s self-similarity, then there is a disagreement in how models respond and the epistemic uncertainty is high. In Appendix A.1, we provide a detailed interpretation of $D(\omega \,\|\, \omega^*)$ as a one-sided kernel discrepancy, establish its connections to variational inference, and show that it is upper bounded by total variation distance under mild conditions.

**Desired Properties for $\Omega$.** Because the divergence $D(\omega \,\|\, \omega^*)$ is evaluated against samples drawn from a *surrogate* distribution of models $\Omega$, its fidelity hinges on how well that ensemble of models approximates the (inaccessible) optimal distribution $p(\cdot \mid x; \omega^*)$. Three criteria follow from the definition in Eq. 3:

  (i) **Support richness.** $\Omega$ covers distinct yet plausible interpretations of models, rather than a narrow subset; otherwise the cross–similarity term in $D$ may be artificially high, and $D$ underestimates EU when predictions of models in $\Omega$ are different from $\omega$'s predictions.
 (ii) **Non-collapsing diversity.** If all members of $\Omega$ are nearly identical (e.g. noise-perturbed versions of the same model), the ensemble average would be too close to $\omega$, hence the cross-model similarity term will be close to self-similarity and $D$ may be small, even when the candidate predictor $P_\omega$ is *mis-specified*.
(iii) **Calibrated weighting.** Let $P_\Omega$ denote the mixing measure over models. For Eq. 3 to approach the ideal $p(y \mid x)$, each model should be weighted in proportion to its posterior credibility (e.g. uniform weights are appropriate only when validation risks are comparable).

**Achieving Properties via Cross-Family Models.** A practical way to satisfy these criteria is to construct the surrogate ensemble $\Omega$ from models of similar architecture and scale, likely trained on overlapping or similar pre-training datasets. Specifically, we populate $\Omega$ with 7–9B Transformer-based models that share the *same architecture class* but are trained by *different vendors*. This setup ensures (i) *support richness*, as models differ in data pipelines, initializations, and alignment protocols, which results in diverse but plausible responses for the same input, that cover the ground-truth response set. These independently trained models also exhibit (ii) *non-collapsing diversity*, as their differences arise from different design choices, rather than noise-perturbed versions of a single model. Finally, because these models achieve similar validation performance, we adopt uniform weights in $P_\Omega$, which satisfies the *calibrated weighting* (iii) requirement. Section 4 specifies the exact models used.

**Total Predictive Uncertainty.** We make the standard assumption that total predictive uncertainty can be obtained by adding aleatoric and epistemic predictive uncertainties [20]. Thus, we define $U_{\text{total}}(x; \omega)$ as:

$$
\begin{aligned}
U_{\text{total}}(x; \omega) &= U_{\text{aleatoric}}(x; \omega) + U_{\text{epistemic}}(x; \omega) \\
&= \mathbb{E}_{\tilde{\omega} \sim P_\Omega} \mathbb{E}_{r_1^\omega \sim p(\cdot|x,\omega)} \mathbb{E}_{q_2^\omega \sim p(\cdot|x,\tilde{\omega})}\big[1 - s(r_1^\omega, q_2^{\tilde{\omega}})\big].
\end{aligned}
\tag{4}
$$

### 3.3 Empirical Estimates of Uncertainty Metrics

For a given input prompt $x$, we call the model whose uncertainty is being estimated the *reference model* $\omega$, and denote the set of models used to compute epistemic uncertainty with respect to the reference as the *auxiliary model set* $\Omega$. Throughout the paper, we mainly focus on *Cross-family auxiliary models*: We estimate epistemic uncertainty by computing response divergence across an auxiliary set of models. To estimate uncertainty in practice, we proceed as follows:

  1. Sample $n$ responses from each model $\omega_i \in \Omega$, and denote the set of responses from $\omega$ as $R' = \{r'_1, r'_2, \ldots, r'_n\}$ and from $\omega_i$ as $R_i = \{r_1^{(i)}, r_2^{(i)}, \ldots, r_n^{(i)}\}$ where $|\Omega| = m$.

2. Approximate Aleatoric, Total, and Epistemic Uncertainty using these sampled responses.

---

**Empirical Uncertainty Metrics**

$$AU = U_{\text{aleatoric}} = 1 - \Big[\sum_{k=1}^{n}\sum_{j=1}^{n} s(r'_k, r'_j)\Big]/n^2$$

$$TU = U_{\text{total}} = 1 - \frac{1}{m}\sum_{i=1}^{m}\Big[\sum_{k=1}^{n}\sum_{j=1}^{n} s(r'_k, r_j^{(i)})\Big]/n^2$$

$$EU = U_{\text{epistemic}} = U_{\text{total}} - U_{\text{aleatoric}}$$

---

Note that we assign uniform weights to different models in the auxiliary set, as we choose to use models of similar capabilities, and our estimate of AU is similar to the one from [39]. Also, observe that our evaluation shows that we can keep the overall number of sampled responses at the magnitude used by self-consistency based methods while improving over those. More concretely, we choose $n = \frac{n'}{m}$, when comparing to AU as a standalone metric, where $n'$ is the number of samples used for the latter.

## 4   Experimental Setup

**Models.** In the main experiments, we primarily focus on five instruction-tuned language models with approximately 7–9B parameters: `Gemma-2-9B-It` [59], `Granite-3.0-8B-Instruct` [17], `Llama-3.1-8B-Instruct` [18], `Mistral-7B-Instruct-v0.3` [23], `Qwen2.5-7B-Instruct` [67]. We compute the uncertainty measures from Section 3.3 and consider the models mentioned above as the set of auxiliary models. In Appendix A.5, we also consider larger reference models. Unless otherwise noted, we compute TU by sampling 2 responses from each of the 5 models and similarly compute AU using 10 samples to keep the sampling budget the same across the two metrics.

**Datasets.** Our experiments cover a broad range of long-form generation tasks spanning question answering (QA), math reasoning, translation, and summarization. For QA, we include `AmbigQA` [45] (open-domain QA with both ambiguous and unambiguous questions), `NQ-open` [33] (closed-book QA derived from real user queries), `HotpotQA` [68] (multi-hop QA requiring reasoning over multiple supporting documents), `CoQA` [51] (conversational QA with multiple turns), `QASPER` [10] (fact-based QA over long scientific papers), `TriviaQA` [26] (QA based on trivia-style questions), and `TruthfulQA` [37] (QA to evaluate common misconceptions in models). For math reasoning, we use `GSM8K` [8] with chain-of-thought prompting. For language generation, we evaluate on the German-to-English translation dataset `WMT16-de-en` [4] and the summarization benchmark `XSum` [47]. We additionally include `SimpleQA` [64], a factuality QA benchmark, with model responses generated by `GPT-4o` [21] and `Claude 3.7 Sonnet` [7]. Finally, we adapt tasks from the BBH multiple-choice benchmark [58] to long-form format and add those evaluations to Appendix A.7.

**Evaluation.** Correctness is defined per input-response pair using `Meta-Llama-3-70B-Instruct` as judge (Appendix A.10). Note that, in the context of uncertainty estimation, LM-as-a-judge correctness evaluation has recently been shown to be the most reliable among the existing methods [52]. Following prior work [39, 31, 3], we evaluate the quality of uncertainty by quantifying how well uncertainty scores separate correct from incorrect generations, using Area Under the ROC Curve (AUROC). Formally, AUROC corresponds to the probability that a randomly chosen incorrect response receives a higher uncertainty score than a randomly chosen correct one.

We also evaluate effectiveness in terms of selective prediction using Risk–Coverage Curves [46], which measures how the error rate changes as uncertain responses are rejected. We further report standard summary metrics such as accuracy at 90% and 80% coverage (C@90 and C@80), and the Area Under Risk-Coverage Curve (AURC), where lower is better.

**Baselines.** As our primary aleatoric baseline, we use Lin et al. [39]'s implementation of Aleatoric Uncertainty, which is in practice similar to Semantic Entropy [31], and has been shown to perform well in recent benchmarks and surveys [12, 61]. In our evaluation, we denote this baseline as Aleatoric or AU. We also experiment with noise-perturbed models (i.e., instead of models from different model families), similar to the approach of Liu et al. [41], see details in Appendix A.5.

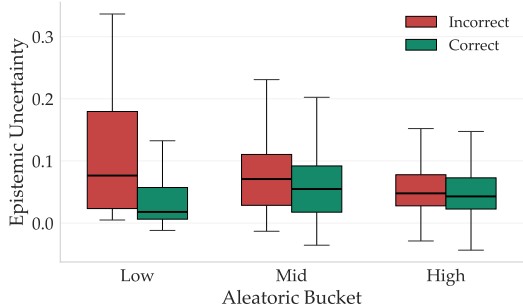 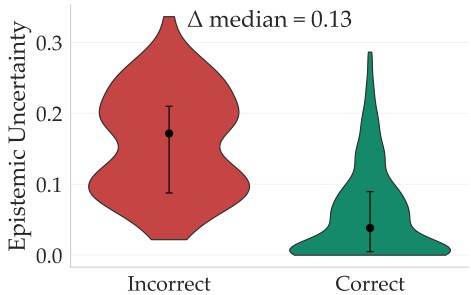

(a) We stratify examples by AU (low, mid, high; 33% each) and compare the distribution of EU for correct and incorrect generations. Incorrect responses show higher EU in the low-AU regime, but this separation weakens as AU increases.

(b) We isolate the most confident samples (lowest 5% AU) and find that incorrect generations have significantly higher EU than correct ones, which shows that EU effectively flags confidently wrong predictions.

Figure 2: Based on the distribution of EU across samples with different AU values, we find that EU separates incorrect from correct most strongly when AU is low.

## 5 Results

### 5.1 Epistemic Uncertainty Flags Confident Failures of Aleatoric Uncertainty

Language models are often applied to heterogeneous tasks, where model confidence does not always align with correctness [69]. To simulate such a setting, we construct an aggregated dataset by combining all datasets mentioned in Section 4, and analyze uncertainty trends on this pooled distribution. We are particularly interested in identifying failure modes of AU.

In Figure 2a, we stratify examples by AU (low, mid, high) and compare EU across correct and incorrect responses. In the low-AU regime, incorrect responses exhibit higher EU than correct ones, which shows that EU is discriminative when aleatoric scores are overconfident. This separation diminishes in higher AU buckets, where both response groups become more uncertain.

To more directly target this failure mode, we isolate the lowest 5% of AU scores and analyze EU by correctness. (Figure 2b). EU remains significantly higher for incorrect generations, which confirms that epistemic uncertainty flags confidently wrong outputs that aleatoric scores alone miss, which supports our hypothesis of the complementary nature of scores in this particular AU region.

This result contrasts with prior work, which treats low-AU predictions as reliable and only incorporates cross-model comparisons when AU exceeds a threshold [66, 6]. Our findings reveal that this assumption overlooks a critical failure mode: confidently wrong predictions with low AU. On the other hand, our observations validate findings about models being overconfident on `HotpotQA` [49] in that incorporating EU yields large improvements on this dataset (see Figure 4 in Section 5.3). Per-dataset results are provided in Figure 18 in Appendix A.8.

### 5.2 Epistemic Uncertainty, Agreement, and Diversity

We ask *when* similarity-based EU is most informative. To this end, we focus on the correctness of responses of different models and consider two metrics: *Jaccard Agreement* (or *Redundancy*) ($J$), which measures the overlap between predicted correct responses of the auxiliary models, used to quantify how redundant or similar different predictions are; and *Oracle Coverage Gain* (also *Complementarity*) ($G$), the additional coverage (i.e., improvement in accuracy) obtained by an oracle that always chooses the correct model per example, over the best performing model. Exact definitions can be found in Appendix A.2.

**Epistemic uncertainty does not always coincide with inter-model disagreement.** Figure 3 plots EU AUROC against the dataset-level statistics $J$ and $G$. We observe a positive correlation with redundancy ($r = +0.72$, $p = 0.03$) and a negative correlation with complementarity ($r = -0.72$, $p = 0.03$), which is the opposite of the naive intuition that "more disagreement $\Rightarrow$ higher epistemic utility."

The explanation lies in how EU is constructed: it grows with the *divergence of generated answers*, which arises in two distinct cases: (i) true EU on intrinsically hard questions where models do not know the answer, and (ii) the existence of many semantically different but correct responses (response noise). In complementary datasets (large $G$), each model specializes on different niches

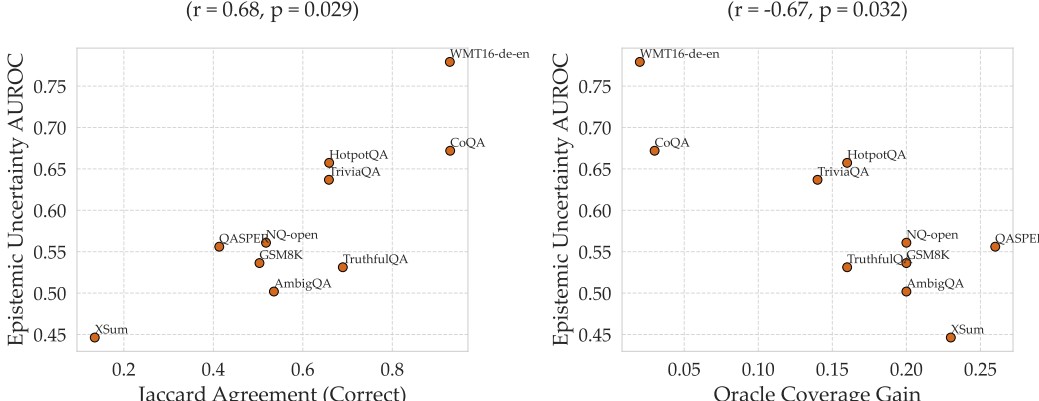

Figure 3: Epistemic–uncertainty AUROC versus dataset–level redundancy ($J$) and complementarity ($G$). Higher AUROC indicates better discrimination between correct and incorrect answers by EU.

and consequently, EU is large even on questions that an individual model answers correctly, because models in the auxiliary set return alternative (wrong) responses. In such cases, the misalignment with correctness drives AUROC down. Conversely, in redundant datasets (high $J$, low $G$), models converge to similar responses when correct (EU low) and, what we expect to be the usual case, still diverge when collectively wrong (EU high), which gives a well-separated score and high AUROC. These observations characterize the cases in which our current EU estimator is effective: tasks with a single (or near-unique) correct answer, where models phrase that answer similarly yet generate diverse alternatives on the harder, unanswered inputs.

For example, `WMT16-de-en` and `CoQA` occupy the high-$J$, low-$G$ corner of Figure 3; all models score above $> 90\%$ accuracy, so predictions are largely redundant and EU achieves its strongest discrimination. At the opposite extreme, `XSum` combines low accuracy with the largest $G$: models succeed on different inputs and can express many valid summaries, which inflates EU without improving ranking and thus lowering AUROC. Datasets such as `HotpotQA` and `TriviaQA` sit mid-range on both axes and have enough redundancy to suppress noise, but have sufficient diversity to expose disagreement and consequently produce the large TU gains in Figure 4.

## 5.3   Total Uncertainty Improves Correctness Calibration

Figure 4 reports the AUROC between negative uncertainty and correctness across datasets, and averaged over five 7–9B instruction-tuned models mentioned in Section 4. TU consistently improves over AU on all benchmarks on average. The largest gains occur on `HotpotQA` (+0.15), `CoQA` (+0.14), and `WMT16-de-en` (+0.13), where models either disagree on complex multi-hop reasoning (`HotpotQA`) or achieve high overall accuracy (`CoQA`, `WMT16-de-en`), which allows EU to capture remaining errors.

Moderate improvements are observed on `TriviaQA`, and `NQ-open`, which exhibit a balance of response *redundancy* and *complementarity*. In contrast, gains are more limited on `TruthfulQA`, `GSM8K` (with chain-of-thought), and `QASPER`, where the presence of multiple valid or stylistically diverse answers weakens the alignment between TU and correctness. These results align with the patterns described in Section 5.2: TU and EU are most effective when correct answers are uniquely phrased and shared across models, while incorrect predictions remain diverse.

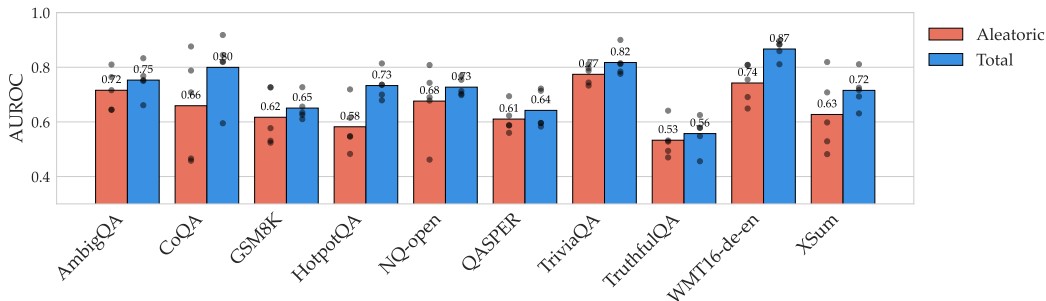

Figure 4: Under matched sample budgets, TU (AU+EU) consistently shows higher AUROC than AU across datasets, with the largest gains on `HotpotQA`, `CoQA`, and `WMT16-de-en` ($\Delta$ >0.10). Bars show means over five 7–9B reference models with per-dataset dots.

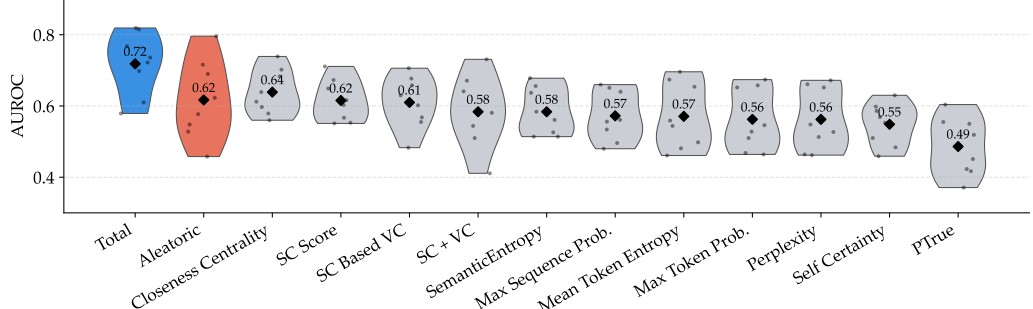

Figure 5: Using `Mistral-7B-Instruct-v0.3` as the reference model, TU attains the best mean AUROC (0.72), and outperforms the strongest baseline (closeness centrality, 0.64) across almost all datasets. Per-task results appear in Table 3 in Appendix A.4.

We also find that TU estimates consistently improve AUROC as compared to AU in `GPT-4o` (0.70 vs. 0.59), and `Claude 3.7 Sonnet` (0.58 vs. 0.53) on `SimpleQA` as shown in Figure 1. Figure 11 in Appendix A.4 shows the ROC curves on the combination of all datasets, where the relative ranking of data points across the whole dataset determines performance, and Table 2 reports AUROC per model-dataset pair. We show that the improvement over AU is maintained in individual datasets, and in the combination of all datasets.

**Comparison to Baselines.** We compare TU against a number of baselines: Mean Token Entropy [13], Maximum Token Probability [12], Maximum Sequence Probability [12], Perplexity [13], PTrue [27], Self-Certainty [28], Semantic Entropy [31], SC (Self-Consistency) Score [63, 44], Closeness Centrality, SC + VC (Verbalized confidence), and SC based CV [24]. Figure 5 shows results for `Mistral-7B-Instruct-v0.3` as the reference model across benchmarks and baselines. Table 3 in Appendix A.4 shows that TU outperforms the strongest baseline across almost all benchmarks.

**Ablations.** We further ablate the size of the reference model in Appendix A.5, and show that even in scenarios where the reference model is larger (and has higher accuracy) than models in the auxiliary set, TU still achieves higher AUROC than AU. Furthermore, we show that AUROC improves with a larger number of sampled responses in Appendix A.6.

## 5.4 Total Uncertainty Improves Selective Abstention

To evaluate whether uncertainty effectively distinguishes reliable responses from potential errors, we consider selective prediction, where models are allowed to abstain from answering when uncertain.

**Risk-Coverage Tradeoff.** Figure 7(a) shows the Risk-coverage curve for aleatoric and total uncertainty, aggregated across all models mentioned in Section 4. Across all coverage levels and datasets, total uncertainty achieves the lowest risk, with a single exception. This suggests that total uncertainty more effectively identifies unreliable predictions in comparison to AU.

**Selective Accuracy and AURC.** To quantify this effect more precisely, Table 1 reports selective accuracy at fixed coverage levels (C@90, C@80) and AURC (area under the risk–coverage curve) across benchmarks and averaged over different models. In nearly all cases, total uncertainty achieves

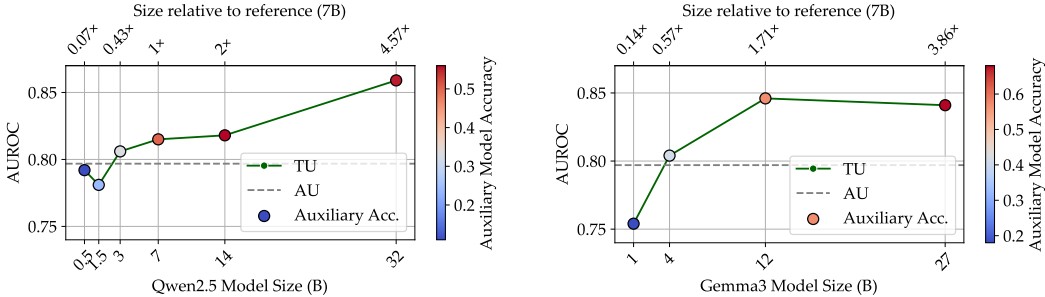

Figure 6: We keep the reference model fixed as `mistral-7B`, and vary the size of the **single auxiliary model**. TU achieves higher AUROC in comparison to AU, even in cases where the size of the auxiliary model is lower than (×0.43) or roughly the same (×1) as the reference model. The improvements are more significant with larger and more capable the auxiliary models on `TriviaQA`.

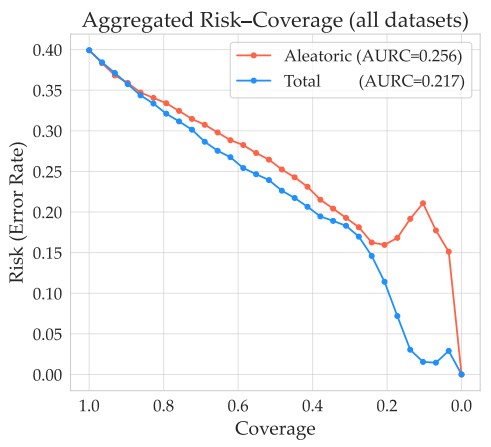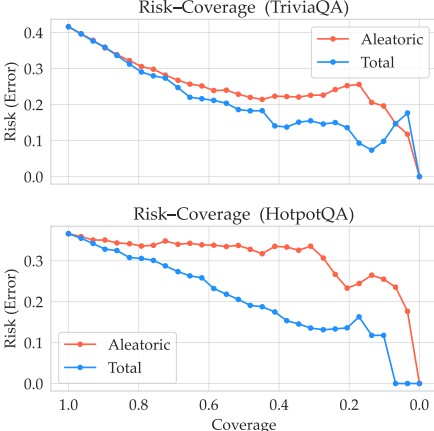

Figure 7: Risk–coverage analysis shows that TU consistently improves selective prediction across datasets and in aggregate.

higher selective accuracy and in all cases lower AURC compared to aleatoric and EU alone. For example, on `HotpotQA` and `XSum`, total uncertainty improves C@90 by over 1.5 points and reduces area under the risk–coverage curve (AURC ↓) by over 20%. These results confirm that TU yields better abstention behavior than AU or EU alone.

Table 1: Selective question answering performance of different uncertainty estimates.

| Dataset | C@90 (%) | | | C@80 (%) | | | AURC ↓ | | |
|---|---|---|---|---|---|---|---|---|---|
| | Aleatoric | Epistemic | Total | Aleatoric | Epistemic | Total | Aleatoric | Epistemic | Total |
| `AmbigQA` | 56.0 | 52.4 | **56.2** | 59.5 | 53.2 | **60.2** | 0.325 | 0.456 | **0.278** |
| `CoQA` | 96.0 | **96.9** | 96.0 | 96.5 | 97.5 | **97.8** | 0.026 | 0.016 | **0.011** |
| `GSM8K` | 54.0 | **54.7** | 53.3 | **54.8** | 54.0 | 54.2 | 0.391 | 0.441 | **0.335** |
| `HotpotQA` | 64.9 | 66.2 | **66.9** | 66.2 | 67.2 | **69.2** | 0.304 | 0.257 | **0.206** |
| `NQ-open` | **53.8** | 51.3 | 53.1 | 57.2 | 52.8 | **57.5** | 0.388 | 0.484 | **0.323** |
| `QASPER` | 38.0 | 36.9 | **39.1** | 39.5 | 37.5 | **40.8** | 0.533 | 0.602 | **0.503** |
| `TriviaQA` | **64.0** | 60.4 | **64.0** | 69.0 | 61.8 | **70.2** | 0.254 | 0.343 | **0.208** |
| `TruthfulQA` | **74.4** | 73.8 | 74.2 | **75.0** | **75.0** | 73.0 | 0.220 | 0.251 | **0.195** |
| `WMT16-de-en` | 96.2 | **96.4** | 96.0 | 97.2 | 96.2 | **98.5** | 0.028 | 0.027 | **0.010** |
| `XSum` | 24.4 | 24.9 | **25.6** | 26.5 | 22.0 | **27.3** | 0.681 | 0.759 | **0.609** |

## 6 Conclusions

We propose that aleatoric and epistemic uncertainty capture complementary failure modes of language models: self-consistency methods reveal data ambiguity, while semantic disagreement across models uncovers uncertainty arising from model limitations. We operationalize this view by estimating TU as the combination of intra-model entropy and inter-model semantic divergence, using only black-box access to model outputs. We show that this combination effectively outperforms self-consistency-based methods across a wide range of models and datasets in terms of different metrics. While this approach requires access to multiple comparable models, it reveals the limits of single-model uncertainty scores and offers a practical path toward more comprehensive uncertainty estimation.

**Limitations.** Our method relies on response-level semantic similarity, which may underperform in tasks with many semantically distinct but correct answers, e.g., open-ended generation or QA tasks where there are multiple distinct correct answers. In such cases, disagreement does not necessarily reflect uncertainty. Additionally, we focus on a specific form of AU; how to best combine our EU estimator with other AU and EU estimators (e.g., token-level or logit-based methods) is left for future work. Moreover, the performance of TU depends on the model ensemble: If all surrogate models share similar pre-training data or architectural biases, cross-model disagreement can underestimate true epistemic uncertainty. We examine this homogeneous-failure scenario in detail in Section 5.2. Finally, our evaluation hinges on a correctness judge; improvements in judge reliability will propagate to more precise AUROC and selective-risk estimates.

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

# A  Appendix

## A.1  Theoretical Interpretations of Epistemic Uncertainty

*Kernel and variational interpretation of $D(\omega \| \omega^*)$.* Assume the similarity function $s(\cdot, \cdot)$ is a symmetric positive definite kernel $k$. Denote the predictive distributions by $P_\Omega := p(\cdot \mid x; \omega)$ and $P_{\omega^*}$. Their kernel mean embeddings in the reproducing kernel Hilbert space (RKHS) $\mathcal{H}_k$ are

$$\mu_\omega = \mathbb{E}_{r \sim P_\Omega}\big[k(r, \cdot)\big], \qquad \mu_{\omega^*} = \mathbb{E}_{q \sim P_{\omega^*}}\big[k(q, \cdot)\big].$$

Using the reproducing property $\langle k(r, \cdot), k(r', \cdot) \rangle_{\mathcal{H}_k} = k(r, r')$, the divergence in Eq. 2 can be rewritten exactly as:

$$D(\omega \| \omega^*) = \langle \mu_\omega, \mu_\omega \rangle_{\mathcal{H}_k} - \langle \mu_\omega, \mu_{\omega^*} \rangle_{\mathcal{H}_k}. \tag{5}$$

Eq. 5 is the first two terms of the squared maximum mean discrepancy (MMD):

$$\mathrm{MMD}^2(P_\Omega, P_{\omega^*}) = \|\mu_\omega - \mu_{\omega^*}\|_{\mathcal{H}_k}^2 = \underbrace{\|\mu_\omega\|_{\mathcal{H}_k}^2 - \langle \mu_\omega, \mu_{\omega^*} \rangle_{\mathcal{H}_k}}_{D(\omega \| \omega^*)} + \|\mu_{\omega^*}\|_{\mathcal{H}_k}^2 - \langle \mu_\omega, \mu_{\omega^*} \rangle_{\mathcal{H}_k}.$$

Thus $D(\omega \| \omega^*)$ is a *one-sided kernel discrepancy*: it measures how much the model's self-agreement exceeds its agreement with the ideal predictor, and it vanishes if and only if $\mu_\omega = \mu_{\omega^*}$ (and, for characteristic kernels, iff $P_\Omega = P_{\omega^*}$).

*Variational-gap Interpretation.* Write classical KL as $\mathrm{KL}(P_\Omega \| P_{\omega^*}) = \mathrm{CE} - \mathrm{Ent}$, where $\mathrm{Ent} = \mathbb{E}_{r \sim P_\Omega}[-\log p(r \mid x; \omega)]$ and $\mathrm{CE} = \mathbb{E}_{r \sim P_\Omega}[-\log p(r \mid x)]$. Replacing $-\log$ with $-k$ yields

$$D(\omega \| \omega^*) = \underbrace{\mathbb{E}_{r, r' \sim P_\Omega}[k(r, r')]}_{\text{"negative kernel-entropy"}} - \underbrace{\mathbb{E}_{r \sim P_\Omega, q \sim P_{\omega^*}}[k(r, q)]}_{\text{"kernel cross-entropy"}},$$

so $D$ is the *semantic variational gap* between the model and the ideal distribution under the geometry induced by $k$. Minimizing $D$ therefore projects $P_\Omega$ toward $P_{\omega^*}$ in RKHS while simultaneously penalizing model variability in $P_\Omega$.

**Lemma 1** (Bound on Kernel-Based Divergence). *Let $P_\omega = p(\cdot \mid x, \omega)$ and $P_{\omega^*} = p(\cdot \mid x, \omega^*)$ be the predictive distributions of two language models. Let $k(\cdot, \cdot)$ be a symmetric, bounded, positive-definite kernel such that $0 \le k(r, r') \le 1$ for all $r, r'$. Define the kernel-based divergence*

$$D(\omega \| \omega^*) := \mathbb{E}_{r, r' \sim P_\omega}[k(r, r')] - \mathbb{E}_{r \sim P_\omega, q \sim P_{\omega^*}}[k(r, q)].$$

*Then $D$ is bounded above in absolute value by the total variation distance between $P_\omega$ and $P_{\omega^*}$:*

$$|D(\omega \| \omega^*)| \le \mathrm{TVD}(P_\omega, P_{\omega^*}),$$

*where*

$$\mathrm{TVD}(P_\omega, P_{\omega^*}) := \frac{1}{2} \int |p_\omega(z) - p_{\omega^*}(z)| dz.$$

*Furthermore, if the RKHS norm of the embeddings are equal, i.e., $\|\mu_\omega\| = \|\mu_{\omega^*}\|$, then $D(\omega \| \omega^*) = 0$ implies $\mu_\omega = \mu_{\omega^*}$. If $k$ is characteristic, this further implies $P_\omega = P_{\omega^*}$.*

*Proof.* Define the kernel-smoothed function $f(z) := \mathbb{E}_{r \sim P_\omega}[k(r, z)]$. Then we can write

$$D(\omega \| \omega^*) = \mathbb{E}_{z \sim P_\omega}[f(z)] - \mathbb{E}_{z \sim P_{\omega^*}}[f(z)].$$

By the definition of total variation distance and the fact that $0 \le f(z) \le 1$ for all $z$,

$$|D(\omega \| \omega^*)| \le \sup_{\|f\|_\infty \le 1} |\mathbb{E}_{P_\omega}[f] - \mathbb{E}_{P_{\omega^*}}[f]| = \mathrm{TVD}(P_\omega, P_{\omega^*}).$$

For the second part, note that $D(\omega \| \omega^*) = \|\mu_\omega\|^2 - \langle \mu_\omega, \mu_{\omega^*} \rangle$. If $\|\mu_\omega\| = \|\mu_{\omega^*}\|$ and $D = 0$, then by Cauchy–Schwarz equality, we must have $\mu_\omega = \mu_{\omega^*}$. If the kernel is characteristic, then $\mu_\omega = \mu_{\omega^*}$ implies $P_\omega = P_{\omega^*}$. $\qquad\square$

This result implies that $D(\omega \,\|\, \omega^*)$ provides a lower bound on distributional mismatch, and thus serves as a tractable proxy for epistemic uncertainty: it is provably small only when the model's predictive distribution aligns with that of the ensemble under the kernel geometry.

Under desired conditions mentioned in 3.2, and with a bounded characteristic kernel, $D(\omega \,\|\, \omega^*) = 0$ if, and only if, the predictive distribution $P_\omega$ is close to the model set consensus as shown above. Consequently $D$ (i) flags cases where the model's intra-model similarity is high (AU is low) while its agreement with the ensemble is low, (ii) rewards calibrated agreement by attaining its minimum only within the support of the ensemble, and (iii) remains sample-efficient, as it only requires $\mathcal{O}(n^2)$ kernel evaluations on $n$ generated responses per model, which is cheaper than KL divergence evaluations or other approximations of EU.

## A.2 Agreement and Coverage Metrics.

For every dataset we form the binary correctness matrix $C_{ij}$ which is the correctness of response $r_i^{(j)}$ sampled from model $j$ to input $i$, and compute two coarse descriptors of cross-model behaviour:

- **Jaccard redundancy** $J$ – The mean pairwise Jaccard index of the sets of correctly answered examples:

$$ J = \frac{2}{M(M-1)} \sum_{1 \le m < k \le M} \frac{|S_m \cap S_k|}{|S_m \cup S_k|}, \quad S_m = \{i : C_{im} = 1\} $$

High $J$ means models succeed on the same inputs.

- **Oracle-gain diversity** $G$ – the additional coverage obtained by an oracle that chooses the correct model per example:

$$ G = A_{\text{oracle}} - \max_m A_m = \frac{1}{N} \sum_{i=1}^{N} \not\Vdash \left[ \sum_{j=1}^{M} C_{ij} > 0 \right] - \max_m \left( \frac{1}{N} \sum_{i=1}^{N} C_{im} \right) $$

High $G$ indicates that different models get different examples right.

## A.3 Additional Experimental Setup

All experiments were conducted on two NVIDIA A100 80GB GPUs. We use the `lm-evaluation-harness` [57] codebase to generate responses from each model, and evaluate correctness using a local vLLM [34] server hosting `Meta-Llama-3-70B-Instruct` [18] as the judge model. Following prior work [39], we compute correctness using only the first sampled response from each model. All evaluation is conducted in inference-only mode; no training or fine-tuning is performed.

For each dataset, we sample 10 responses per model for the first 100 prompts. AU is computed using all 10 responses. To match the sample budget with TU, the experiments in Section 5.3 and Section 5.4 report results using only 2 responses per model for computing both TU and epistemic uncertainty. We use a temperature of 0.7 and top-p of 0.9 across all generations. For `SimpleQA`, model outputs are obtained from the OpenAI and Anthropic APIs.

Semantic similarity between responses is measured using cosine distance over `sentence-T5-xl` [19, 48] embeddings. For datasets not originally supported by `lm-eval-harness`, we follow its prompt formatting conventions and include code for these additions in the supplementary material.

## A.4 Additional Results on Total Uncertainty

Figure 8 reports the AUROC of aleatoric and total uncertainty across all model–dataset pairs, and Figure 9 shows the corresponding improvements from using TU over AU. Total uncertainty consistently improves correctness discrimination in nearly all cases, with the largest per-instance gains observed on `HotpotQA`, a benchmark known for complex multi-hop reasoning.

**AUROC Results Per Model and Dataset.** Figure 8 and Table 2 show AUROC of TU and AU in all model-dataset combinations, and Figure 9 shows the improvement of total uncertainty over AU in AUROC. TU improves AUROC in nearly all model–dataset combinations, with the largest gains

observed in `HotpotQA`. Model performance substantially affects the magnitude of improvement. On datasets such as `XSum`, where overall model accuracy is low, TU yields large improvements for weaker models but occasionally underperforms for the strongest ones (e.g., `Llama` and `Qwen`), potentially due to disagreement with less reliable auxiliary models. A similar pattern holds on `GSM8K`, where gains are concentrated among lower-performing models, while others benefit less.

In contrast, `WMT16-de-en` and `CoQA` show limited gains, likely due to the high baseline accuracy ($> 90\%$) of all reference models (see Fig. 10), where AU is already well-calibrated. Notably, TU corrects miscalibrated AU for specific models, such as `Mistral` on `CoQA`, where the base AU is anomalously low. On `TruthfulQA`, which features open-ended questions with diverse valid answers, semantic disagreement does not reliably indicate epistemic uncertainty, which results in weaker improvements.

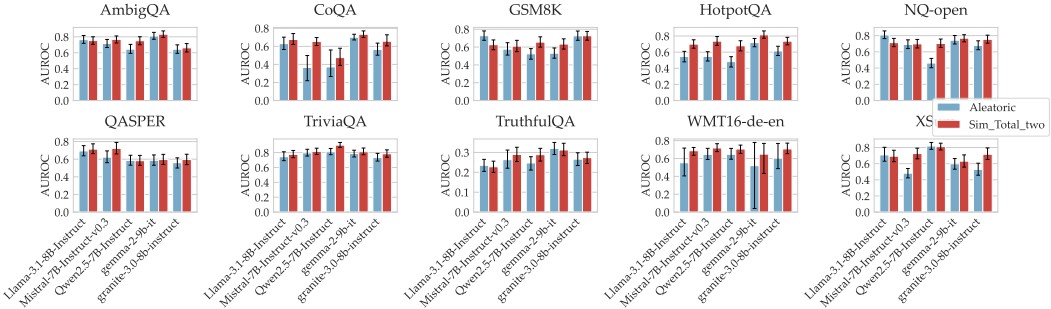

Figure 8: We show AUROC for each model separately to compare aleatoric and total uncertainty. TU consistently yields higher AUROC across models. We subsample 80% of the questions per dataset, 1000 times, to compute AUROC with 5% confidence intervals around the median AUROC value.

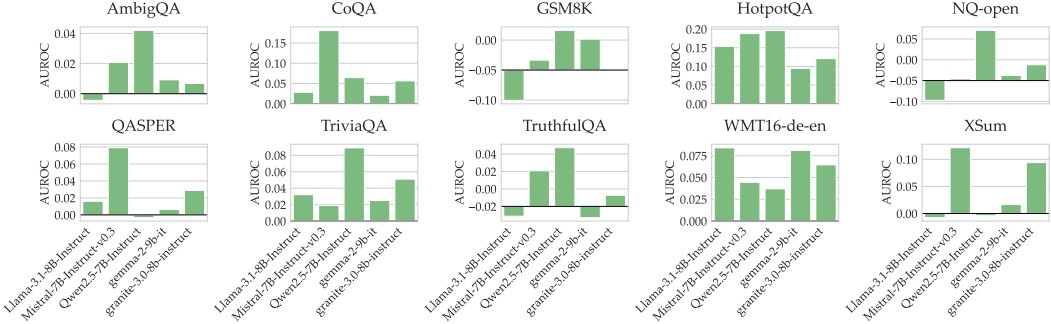

Figure 9: AUROC improvement obtained by adding EU to AU across all samples per dataset, measured as (Total − Aleatoric).

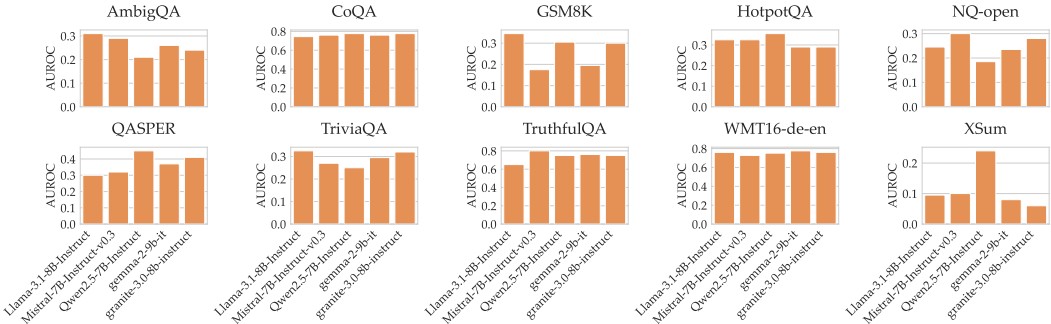

Figure 10: Accuracy per model-dataset pair.

Table 2: Uncertainty AUROC scores across models and benchmarks when different 7B/8B/9B parameter models are used as auxiliary models. Total Uncertainty is better calibrated with correctness than Aleatoric Uncertainty.

| Benchmark | Model | Accuracy | Aleatoric AUROC | Epistemic AUROC | Total AUROC |
|---|---|---|---|---|---|
| AmbigQA | Llama-3.1-8B-Instruct | 0.62 | **0.764** | 0.52 | 0.753 |
| | Mistral-7B-Instruct-v0.3 | 0.58 | 0.716 | 0.506 | **0.768** |
| | Qwen2.5-7B-Instruct | 0.42 | 0.645 | 0.639 | **0.75** |
| | gemma-2-9b-it | 0.52 | 0.81 | 0.447 | **0.833** |
| | granite-3.0-8b-instruct | 0.48 | 0.644 | 0.488 | **0.661** |
| CoQA | Llama-3.1-8B-Instruct | 0.93 | 0.788 | 0.797 | **0.845** |
| | Mistral-7B-Instruct-v0.3 | 0.95 | 0.458 | 0.80 | **0.819** |
| | Qwen2.5-7B-Instruct | 0.97 | 0.466 | 0.567 | **0.595** |
| | gemma-2-9b-it | 0.95 | 0.876 | 0.80 | **0.918** |
| | granite-3.0-8b-instruct | 0.97 | 0.708 | **0.907** | 0.821 |
| GSM8K | Llama-3.1-8B-Instruct | 0.69 | **0.727** | 0.346 | 0.626 |
| | Mistral-7B-Instruct-v0.3 | 0.35 | 0.577 | 0.606 | **0.61** |
| | Qwen2.5-7B-Instruct | 0.61 | 0.524 | 0.623 | **0.656** |
| | gemma-2-9b-it | 0.39 | 0.531 | 0.601 | **0.634** |
| | granite-3.0-8b-instruct | 0.60 | 0.726 | 0.472 | **0.727** |
| HotpotQA | Llama-3.1-8B-Instruct | 0.65 | 0.546 | 0.662 | **0.7** |
| | Mistral-7B-Instruct-v0.3 | 0.65 | 0.548 | 0.682 | **0.736** |
| | Qwen2.5-7B-Instruct | 0.71 | 0.483 | **0.681** | 0.679 |
| | gemma-2-9b-it | 0.58 | 0.719 | 0.608 | **0.814** |
| | granite-3.0-8b-instruct | 0.58 | 0.615 | 0.634 | **0.736** |
| NQ-open | Llama-3.1-8B-Instruct | 0.49 | **0.808** | 0.389 | 0.713 |
| | Mistral-7B-Instruct-v0.3 | 0.60 | 0.69 | 0.501 | **0.698** |
| | Qwen2.5-7B-Instruct | 0.37 | 0.462 | 0.696 | **0.703** |
| | gemma-2-9b-it | 0.47 | 0.743 | 0.538 | **0.768** |
| | granite-3.0-8b-instruct | 0.56 | 0.678 | 0.541 | **0.754** |
| QASPER | Llama-3.1-8B-Instruct | 0.30 | 0.694 | 0.487 | **0.714** |
| | Mistral-7B-Instruct-v0.3 | 0.32 | 0.623 | 0.657 | **0.722** |
| | Qwen2.5-7B-Instruct | 0.45 | **0.587** | 0.492 | 0.583 |
| | gemma-2-9b-it | 0.37 | 0.588 | 0.509 | **0.596** |
| | granite-3.0-8b-instruct | 0.41 | 0.56 | 0.512 | **0.596** |
| TriviaQA | Llama-3.1-8B-Instruct | 0.65 | 0.744 | 0.562 | **0.776** |
| | Mistral-7B-Instruct-v0.3 | 0.54 | 0.796 | 0.552 | **0.815** |
| | Qwen2.5-7B-Instruct | 0.5 | 0.811 | 0.668 | **0.9** |
| | gemma-2-9b-it | 0.59 | 0.787 | 0.645 | **0.812** |
| | granite-3.0-8b-instruct | 0.64 | 0.733 | 0.575 | **0.784** |
| TruthfulQA | Llama-3.1-8B-Instruct | 0.65 | **0.47** | 0.423 | 0.456 |
| | Mistral-7B-Instruct-v0.3 | 0.80 | 0.528 | **0.601** | 0.579 |
| | Qwen2.5-7B-Instruct | 0.75 | 0.494 | **0.584** | 0.578 |
| | gemma-2-9b-it | 0.76 | **0.641** | 0.541 | 0.625 |
| | granite-3.0-8b-instruct | 0.75 | 0.532 | **0.556** | 0.548 |
| WMT16-de-en | Llama-3.1-8B-Instruct | 0.95 | 0.691 | 0.695 | **0.859** |
| | Mistral-7B-Instruct-v0.3 | 0.91 | 0.808 | 0.835 | **0.897** |
| | Qwen2.5-7B-Instruct | 0.94 | 0.809 | 0.832 | **0.883** |
| | gemma-2-9b-it | 0.97 | 0.649 | 0.663 | **0.811** |
| | granite-3.0-8b-instruct | 0.95 | 0.755 | 0.493 | **0.884** |
| XSum | Llama-3.1-8B-Instruct | 0.19 | **0.708** | 0.37 | 0.693 |
| | Mistral-7B-Instruct-v0.3 | 0.20 | 0.482 | 0.651 | **0.725** |
| | Qwen2.5-7B-Instruct | 0.48 | **0.819** | 0.31 | 0.811 |
| | gemma-2-9b-it | 0.16 | 0.598 | 0.565 | **0.631** |
| | granite-3.0-8b-instruct | 0.12 | 0.529 | 0.582 | **0.717** |

**ROC Curves.** Figure 11 shows the ROC curve computed over the pooled set of all model–dataset pairs. TU achieves a higher AUROC (0.746 vs. 0.707), which shows improved seperation between correct and incorrect generations compared to AU alone. Figure 12 presents ROC curves for individual datasets. TU yields consistently better or comparable performance across all tasks, with the largest gains observed on HotpotQA, WMT16-de-en, and CoQA. These improvements align with our earlier findings that TU is most effective on tasks where models are accurate but occasionally confidently wrong.

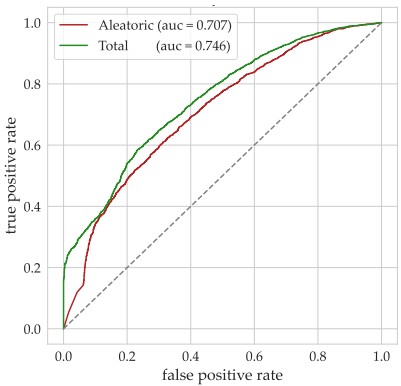

Figure 11: ROC curves between aleatoric and total uncertainty aggregated across all models and datasets. Total uncertainty achieves higher AUROC, indicating better discrimination between correct and incorrect generations.

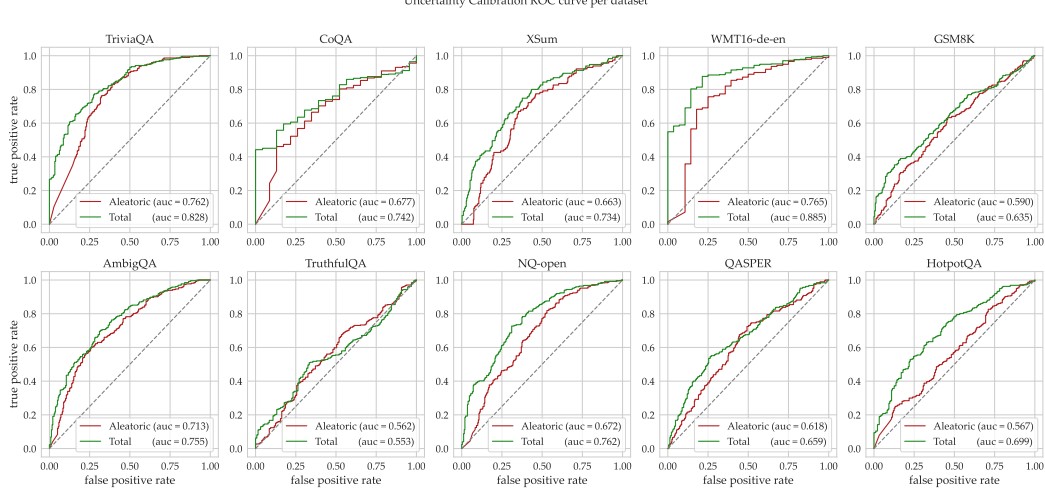

Figure 12: ROC curves comparing aleatoric and total uncertainty across individual datasets. TU achieves higher AUROC across most tasks, particularly on `HotpotQA`, `WMT16-de-en`, and `CoQA`, where models exhibit confident failures.

**Comparison with Baselines.** We compare TU against a number of baselines: Mean Token Entropy [13], Maximum Token Probability [12], Maximum Sequence Probability [12], Perplexity [13], PTrue [27], Self-Certainty [28], Semantic Entropy [31], SC (Self-Consistency) Score [63, 44], Closeness Centrality, SC + VC (Verbalized confidence), and SC based CV [24]. We use implementations from Fadeeva et al. [12], Jiang et al. [24]. AU baselines is the same as [39], but instead of using entailment with Deberta to compute response sequence similarity, we use a `sentence-T5-xl` model. Results are provided in Table 3.

## A.5 The Effect of the Auxiliary Model Set on Total Uncertainty

**Ablating the Reference Model Size.** We test how the quality of total uncertainty estimates depends on the capability of the *reference model*, whose uncertainty we aim to estimate. We fix the auxiliary model set to a pool of four 7-9B models mentioned in 4 that are not from the same family as the reference model, and vary the reference model's architecture and size. Figure 13 reports results on `TriviaQA`, using two model families (Gemma3 [60] and Qwen2.5 [67]) of various sizes. As the size of the reference model increases, both aleatoric and total uncertainty AUROC scores tend to decrease, but total uncertainty has consistently higher AUROC across different model sizes. This holds even when the reference model is substantially stronger than any model in the auxiliary set (e.g., `Qwen2.5-32B` vs. 7-9B peers).

Table 3: Uncertainty AUROC scores across benchmarks for `Mistral-7B` when different 7B/8B/9B parameter models are used as auxiliary models. Total Uncertainty is better calibrated with correctness than Aleatoric Uncertainty and other baselines.

| | AmbigQA | CoQA | GSM8K | HotpotQA | NQ-open | QASPER | TriviaQA | TruthfulQA | **Average** |
|---|---|---|---|---|---|---|---|---|---|
| **Total** | **0.768** | **0.819** | **0.61** | 0.736 | **0.698** | **0.722** | **0.815** | 0.579 | **0.718** |
| **Aleatoric** | 0.716 | 0.458 | 0.577 | 0.548 | 0.69 | 0.623 | 0.796 | 0.528 | 0.617 |
| **Closeness Centrality** | 0.683 | 0.612 | 0.56 | **0.739** | 0.597 | 0.579 | 0.702 | **0.639** | 0.639 |
| **SC Score** | 0.649 | 0.553 | 0.551 | 0.711 | 0.616 | 0.567 | 0.673 | 0.603 | 0.615 |
| **SC Based VC** | 0.658 | 0.483 | 0.555 | 0.677 | 0.63 | 0.568 | 0.706 | 0.602 | 0.61 |
| **SemanticEntropy** | 0.678 | 0.561 | 0.584 | 0.526 | 0.656 | 0.514 | 0.637 | 0.514 | 0.584 |
| **SC + VC** | 0.671 | 0.411 | 0.544 | 0.51 | 0.641 | 0.581 | 0.731 | 0.581 | 0.584 |
| **Max Sequence Prob.** | 0.651 | 0.561 | 0.496 | 0.556 | 0.64 | 0.48 | 0.66 | 0.534 | 0.572 |
| **Mean Token Entropy** | 0.654 | 0.498 | 0.461 | 0.559 | 0.674 | 0.481 | 0.696 | 0.544 | 0.571 |
| **Token Entropy** | 0.654 | 0.502 | 0.465 | 0.558 | 0.672 | 0.481 | 0.69 | 0.542 | 0.57 |
| **Perplexity** | 0.661 | 0.513 | 0.462 | 0.548 | 0.652 | 0.464 | 0.672 | 0.527 | 0.562 |
| **Max Token Prob.** | 0.658 | 0.51 | 0.464 | 0.546 | 0.652 | 0.468 | 0.674 | 0.528 | 0.562 |
| **Self Certainty** | 0.569 | 0.51 | 0.459 | 0.553 | 0.598 | 0.484 | 0.63 | 0.586 | 0.549 |
| **PTrue** | 0.604 | 0.555 | 0.371 | 0.417 | 0.55 | 0.423 | 0.519 | 0.451 | 0.486 |

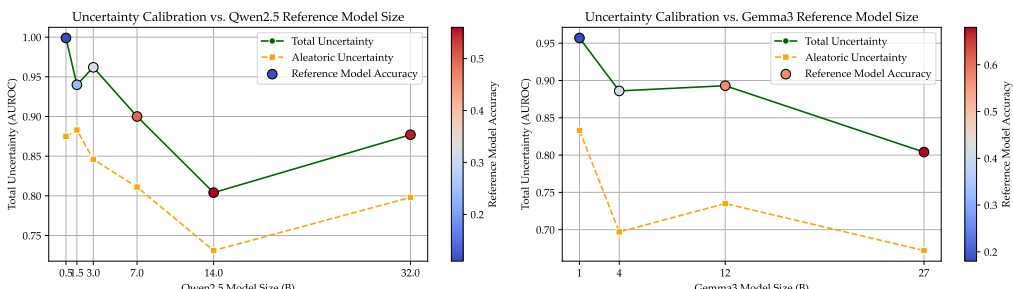

Figure 13: We vary the size of the **reference model** while holding the auxiliary models fixed. TU achieves higher AUROC in comparison to AU acorss different model sizes on `TriviaQA`.

**Noise-perturbed Auxiliary Model Set.** We consider noise-perturbed variants of the reference model itself as auxiliary model set, similar to Liu et al. [41]. Specifically, we apply a perturbation strategy in which we preserve the top-$k$ singular vectors of each linear weight matrix and inject Gaussian noise into the remaining lower-rank subspace. This preserves dominant components of the model while allowing for controlled noise in its response distribution. Figure 14 in the appendix shows that the we sometimes obtain improvement in TU-AU, but it is overall lower than for the more diverse auxiliary model set from Figure 4.

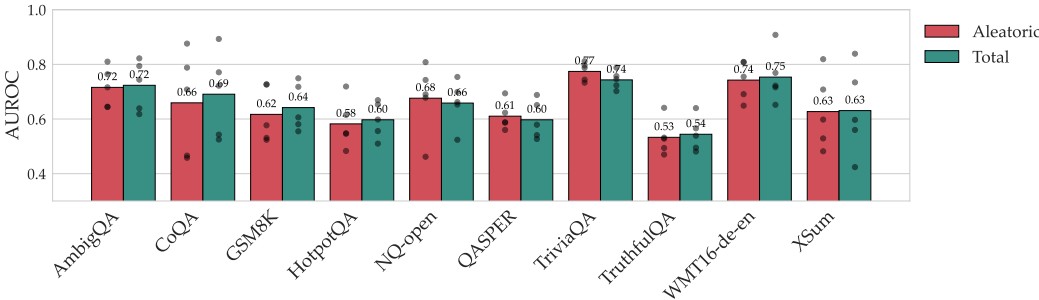

Figure 14: Uncertainty calibration for experiments where auxiliary model set for each model is consisted of multiple noise-perturbed models

## A.6 AUROC Ablations

**Number of Auxiliary Models.** We study how the size of the auxiliary model set affects the quality of total uncertainty estimates. For each reference model, we compute total uncertainty using $n \in \{2, 3, 4, 5\}$ models, where one model is fixed (the reference model) and the remaining $n - 1$ are sampled from other model families. All methods use a fixed number of samples per model.

Figure 15 shows that total uncertainty improves monotonically as the number of auxiliary models increases. This holds across almost all tasks, with the largest gains typically occurring between $n = 2$ and $n = 3$. In addition, we observe that variance across runs decreases as more models are added, which suggests a more calibrated uncertainty score can be achieved from increasing the number of model in the auxiliary set. However, in all datasets, our multi-sample total uncertainty measure outperforms aleatoric uncertainty in AUROC, even when only one auxiliary model is used.

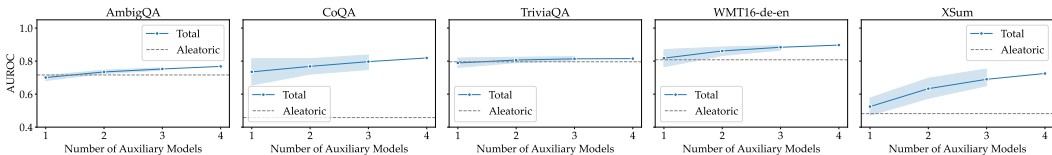

Figure 15: We plot AUROC as a function of the number of auxiliary models used to compute total uncertainty for `Mistral-7B-Instruct-v0.3`. Total uncertainty improves with more models, and variance decreases.

**Number of Samples for Uncertainty Estimation.** We next investigate how the number of response samples per model affects the performance of uncertainty estimates. For each model in the auxiliary set, we vary the number of generations used in total uncertainty computation from 5 to 50, and compare against two baselines for aleatoric uncertainty: one computed using 5 samples and another using 10 samples, matching the regimes used in our main experiments.

As shown in Figure 16, AUROC for total uncertainty usually slightly increases with more samples, with diminishing returns beyond 30 samples in most tasks. Notably, TU consistently outperforms AU baselines across all datasets. These findings also reinforce the practicality of TU even under constrained budgets, as improvements are apparent with as few as 10 samples ($n = 5$ on the x-axis).

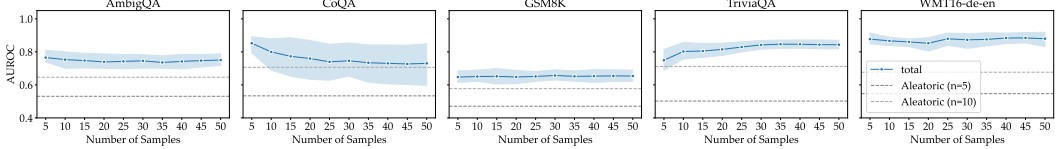

Figure 16: AUROC of total uncertainty as a function of the number of samples per model. Even with a small number of samples, TU outperforms aleatoric baselines (5-sample and 10-sample variants). Gains saturate around 30–40 samples.

### A.7 Additional Results on Multiple-choice QA tasks

To evaluate whether our findings extend beyond open-ended generation, we adapt a subset of tasks from the Big-Bench Hard (BBH) [58] benchmark into a long-form QA format with chain-of-thought answering. Specifically, we consider `Boolean Expressions`, `Disambiguation QA`, and `Word Sorting`, and prompt models to justify their answers rather than selecting from multiple choices directly. We then evaluate uncertainty scores over the full responses using the same semantic similarity pipeline as in our main experiments.

Table 4 reports AUROC scores for both AU and TU across models on these tasks. We observe that TU improves over AU in most cases, with the largest gains appearing when base model performance is low (e.g., Qwen2.5-7B on `Disambiguation QA` and `Boolean Expressions`). These results demonstrate that TU remains effective in identifying incorrect generations even when the task is originally framed as multiple-choice, provided responses are elicited in free-form.

Table 4: Uncertainty AUROC scores across models and benchmarks when different 7B/8B/9B parameter models are used as auxiliary models. Total Uncertainty is better calibrated with correctness than Aleatoric Uncertainty.

| Benchmark | Model | Accuracy | Aleatoric AUROC | Total AUROC |
|---|---|---|---|---|
| **BBH Fewshot Boolean Expressions** | `Llama-3.1-8B-Instruct` | **0.88** | **0.662** | 0.658 |
| | `Mistral-7B-Instruct-v0.3` | **0.84** | **0.746** | 0.735 |
| | `Qwen2.5-7B-Instruct` | **0.53** | 0.744 | **0.909** |
| | `gemma-2-9b-it` | **0.86** | 0.593 | **0.725** |
| | `granite-3.0-8b-instruct` | **0.9** | **0.659** | 0.658 |
| **BBH Fewshot Disambiguation QA** | `Llama-3.1-8B-Instruct` | **0.59** | 0.544 | **0.594** |
| | `Mistral-7B-Instruct-v0.3` | **0.64** | 0.525 | **0.656** |
| | `Qwen2.5-7B-Instruct` | **0.44** | 0.561 | **0.81** |
| | `gemma-2-9b-it` | **0.69** | 0.61 | **0.65** |
| | `granite-3.0-8b-instruct` | **0.62** | 0.486 | **0.562** |
| **BBH Fewshot Word Sorting** | `Llama-3.1-8B-Instruct` | **0.69** | 0.476 | **0.512** |
| | `Mistral-7B-Instruct-v0.3` | **0.77** | **0.529** | 0.429 |
| | `Qwen2.5-7B-Instruct` | **0.44** | 0.587 | **0.645** |
| | `gemma-2-9b-it` | **0.96** | 0.475 | **0.576** |
| | `granite-3.0-8b-instruct` | **0.58** | **0.578** | 0.485 |

## A.8 Epistemic Uncertainty Analysis

Figure 17 disaggregates the trend shown in Figure 2a by model. For all five reference models, incorrect generations in the low-AU regime show consistently higher EU than correct ones, which reaffirms that EU captures confident failures missed by self-consistency. This separation weakens in mid- and high-AU buckets, where both correct and incorrect outputs tend to be more uncertain. The consistency of this pattern across different models highlights the effectiveness of EU in identifying unreliable predictions when AU alone is low. Similarly, Figure 18 shows a similar trend when results are disaggregated by *dataset*.

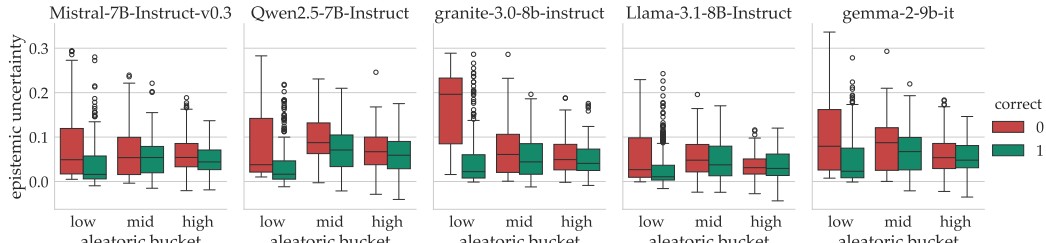

Figure 17: Distribution of EU across different levels of AU and correctness. Across all models, we find that incorrect responses in the low-aleatoric regime are assigned higher EU than correct ones on average.

## A.9 Computational cost

We report full wall-clock measurements to contextualize the overhead introduced by our TU ensemble relative to a self-consistency baseline with the same total sample budget. All runs use **TriviaQA** on a single Nvidia L40S (48 GB) unless noted. Decoding settings and similarity-oracle calls are identical across settings; the observed gap comes from loading additional checkpoints.

| Regime | Setting | Models × samples | Per-model mean (s) | Total wall-clock (s) |
|---|---|---|---|---|
| Single-GPU (sequential) | Single model, 10 samples | $1 \times 7\text{–}9B \times 10$ | $244.1 \pm 58.7$ | 244.1 |
| Single-GPU (sequential) | TU ensemble, 5 models × 2 samples | $5 \times 7\text{–}9B \times 2$ | $78.3 \pm 12.9$ | 391.7 |
| Multi-GPU (parallel, 5×) | TU ensemble, 5 models × 2 samples | $5 \times 7\text{–}9B \times 2$ | $78.3 \pm 12.9$ | $\approx 78$ |

Table 5: **Wall-clock on TriviaQA.** Sequential TU is slower due to streaming four extra checkpoints; peak GPU memory is unchanged. With 5-way parallelism (one GPU per model), wall-clock returns to ~78 s while preserving TU's calibration and abstention gains.

**Takeaways.** 1) *Token-generation and similarity costs are matched* between the self-consistency baseline ($1 \times 10$ samples) and TU ($5 \times 2$). 2) *Sequential overhead is dominated by checkpoint loads*; peak VRAM does not increase. 3) *Parallel execution amortizes loading*: with 5× parallelism, TU

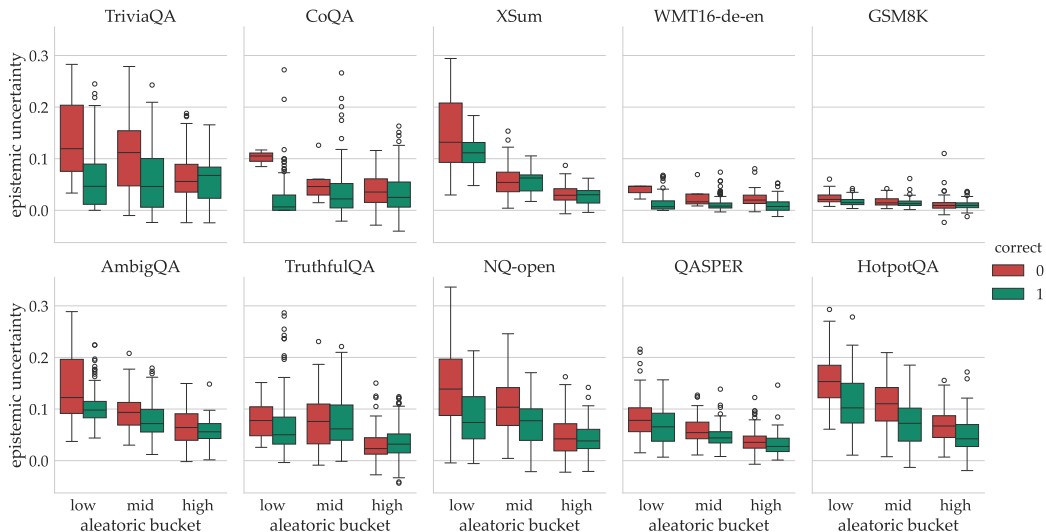

Figure 18: Distribution of EU across different levels of AU and correctness. Across all benchmarks, we find that incorrect responses in the low-aleatoric regime are assigned higher EU than correct ones on average.

matches single-model wall-clock while retaining its AUROC and selective-abstention improvements. In practice, a small ensemble (2–5 auxiliaries, 1–2 samples each) provides most of the TU benefit under common deployment envelopes.

### A.10 Computing Correctness Using an LLM Judge

We compute correctness scores using `Meta-Llama-3-70B-Instruct` deployed via a local `vLLM` server. Each model prediction is evaluated independently against the gold answers using a structured prompt that includes five few-shot examples, held fixed across all evaluations. The prompt instructs the judge to assign a correctness score from the discrete set {0.0, 0.1, ..., 1.0} based on the alignment between the predicted and gold answers, while explicitly ignoring the model's own knowledge.

The judge receives as input: (1) the user-defined task or question, (2) a list of gold answers, and (3) the model-generated answer. It is instructed to output a JSON object containing a numerical score and a justification. The request is submitted using deterministic decoding (`temperature`= 0, `max_tokens` = 20), and we employ up to three retries with truncated context in case of failures due to prompt length.

Correctness is evaluated using the first response generated by each model. The gold answers are passed verbatim, and no normalization is applied to either predictions or references. By default, a prediction is considered correct if its score exceeds 0.5. For tasks where differences should be penalized (e.g., summarization or translation), we increase the threshold to 0.9 (specifically, for `XSum` and `WMT16-de-en`). These thresholds are applied during AUROC and selective prediction evaluations.

**Judge Evaluation.** We assess the reliability of the LLM judge used to score correctness against gold references. First, we perform a cross-judge check (Llama-3-70B-Instruct vs. GPT-4o) and find high agreement on both probabilities and binary labels. In 93/100 cases the judges produced identical correctness probabilities. In 5/100, probabilities differed but mapped to the same binary label under the 0.5 threshold, so AUROC is unchanged. Only 2/100 items were hard conflicts, and manual inspection favored GPT-4o in both.

Second, We manually audited 100 Mistral-generated samples across TriviaQA and TruthfulQA and observed <6% human–judge disagreement. We also compare to rule-based matchers from `lm-evaluation-harness` and find them brittle for free-form outputs, which motivagtes an LLM judge.

**Prompt Format.** We provide the prompt format used for the LLM Judge here.

```
I want you to act as a judge for how well a model did answering a user-defined
task.
You will be provided with a user-defined task that was given to the model, its
golden answer(s), and the model's answer.  The context of the task may not be
given here.  Your task is to judge how correct the model's answer is based on
the golden answer(s), without seeing the context of the task, and then give a
correctness score.  The correctness score should be one of the below numbers:
0.0 (totally wrong), 0.1, 0.2, ..., 1.0 (totally right).  You should also add a
brief justification regarding how the model's answer conforms to or contradicts
the golden answer(s).  Your response must follow the format:
{
"correctness_score":  your_score,
"justification":  your_justification
}
Note that each one of the golden answers is considered correct.  Thus if the
model's answer matches any one of the golden answers, it should be considered
correct.
--
Example 1:
User-defined task -- Sandy bought 1 million Safe Moon tokens.  She has 4
siblings.  She wants to keep half of them to herself and divide the remaining
tokens among her siblings.  After splitting it up, how many more tokens will she
have than any of her siblings?
Golden Answer(s) -- <answer 1> 375000
Model's Answer -- Sandy will have more tokens than any sibling by 3/8 million.
Model Output:
{
"correctness_score":  1.0,
"justification":  "The model's answer of 3/8 million equals 375,000, which
matches the gold answer exactly."
}
--
...  (3 more examples)
--
Target Example:
User-defined task -- [QUESTION]
Golden Answer(s) -- <answer 1> [...]; <answer 2> [...]
Model's Answer -- [MODEL RESPONSE]
Model Output:
{
"correctness_score":  ?,
"justification":  ?
}
```

