# OpenReview forum: "Complementing Self-Consistency with Cross-Model Disagreement for Uncertainty Quantification"
_NeurIPS.cc/2025/Workshop/Reliable_ML — NeurIPS 2025 - Reliable ML Workshop_

### Official Review · Reviewer_pPpJ · 2025-09-19
**Review for AU and Eu in LLMs**

**Rating:** 7
**Confidence:** 3

**Review:**

This paper tackles an important practical problem that aligns well with the workshops theme: when can we trust an LLM’s answer? The idea of combining aleatoric uncertainty (how much a model varies on a fixed input) with an epistemic term (how much it disagrees with other models again conditioning on a fixed input) is clear, sensible, and easy to motivate. I like that the method is training-free and uses only black-box outputs from a few models. The formulation based on response similarity is also easy to implement and explain, though it would help to specify the exact similarity function. The experiments are thoughtful. They use multiple 7–9B instruction tuned models and a diverse set of tasks and they evaluate both ranking (AUROC) and selective prediction (risk coverage, AURC etc.). The main result, that Total Uncertainty (TU) beats aleatoric alone across most datasets, comes through clearly, and there is a convincing analysis of when EU helps the most. A comment about the choice of $\Omega$: In practice, $\Omega$  is a small set of chosen models. It isn’t clear how sensitive TU/EU are to this choice. A sensitivity check or an explanation of why even a few models are enough to satisfy the assumptions for $\Omega$, would be helpful.  Overall though, this is an interesting paper, with a clean theoretical set up to study reliability in LLMs.

---

### Official Review · Reviewer_kukS · 2025-09-20
**A Simple Method to Uncover Confident Failures in LLMs**

**Rating:** 7
**Confidence:** 4

**Review:**

**Summary**
The paper tackles confident but incorrect outputs in LLMs by combining aleatoric uncertainty (AU, intra-model variability) with epistemic uncertainty (EU, cross-model disagreement). The proposed Total Uncertainty (TU = AU + EU) is training-free, uses only black-box outputs, and consistently improves AUROC calibration and selective abstention across diverse tasks and models.

**Strengths**
1. Clear framing of AU vs. EU as complementary failure modes.
2. Simple, practical method leveraging ensembles of existing models.
3. Direct relevance to reliable ML under imperfect/confident errors.

**Weaknesses**
1. Evaluation focuses mainly on AU vs. TU. Other EU estimation methods (token/logit-based Bayesian approaches, iterative prompting, verifier disagreement) are acknowledged but not fully benchmarked, leaving unclear how TU compares in broader landscape.
2. Correctness labels depend on a single LLM-as-a-judge, which may itself introduce biases or errors; robustness of findings to different judges is not analyzed.

**Suggestions**
See weaknesses.